# *N*-Arylation of 3-Formylquinolin-2(1*H*)-ones Using Copper(II)-Catalyzed Chan–Lam Coupling

**DOI:** 10.3390/molecules27238345

**Published:** 2022-11-30

**Authors:** Jhesua Valencia, Oriel A. Sánchez-Velasco, Jorge Saavedra-Olavarría, Patricio Hermosilla-Ibáñez, Edwin G. Pérez, Daniel Insuasty

**Affiliations:** 1Departamento de Química y Biología, División de Ciencias Básicas, Universidad del Norte, Km 5 Vía Puerto Colombia, Barranquilla 081007, Colombia; 2Department of Organic Chemistry, Faculty of Chemistry and Pharmacy, Pontificia Universidad Católica de Chile, Santiago 7820436, Chile; 3Center for the Development of Nanoscience and Nanotechnology (CEDENNA), Materials Chemistry Department, Faculty of Chemistry and Biology, University of Santiago, Chile, Santiago 9170022, Chile

**Keywords:** Chan–Lam coupling, copper(II), *N*-arylation, 3-formylquinolones

## Abstract

3-formyl-2-quinolones have attracted the scientific community’s attention because they are used as versatile building blocks in the synthesis of more complex compounds showing different and attractive biological activities. Using copper-catalyzed Chan–Lam coupling, we synthesized 32 new *N*-aryl-3-formyl-2-quinolone derivatives at 80 °C, in air and using inexpensive phenylboronic acids as arylating agents. 3-formyl-2-quinolones and substituted 3-formyl-2-quinolones can act as substrates, and among the products, the *p*-methyl derivative **9a** was used as a substrate to obtain different derivatives such as alcohol, amine, nitrile, and chalcone.

## 1. Introduction

The *N*-arylation of *N*-heterocyclic compounds has made a great impact on the advance of organic chemistry. This coupling reaction has simplified the generation of new derivatives with a wide range of applications due to their biological [1,2,3], agrochemical, photophysical [4,5,6], and catalytic [7] properties, among others. In this regard, since the last century, the Ullman–Goldberg and Buchwald–Hartwig cross-coupling reactions have played relevant roles using aryl halides for the *N*_heterocycle_–C_aryl_ bond formation [8]. However, these synthetic methods have several drawbacks, such as the use of expensive metal catalysts, toxic solvents, ligands that are not commercially available, or harsh reaction conditions, and, for this reason, new alternatives have been emerging [9,10].

In this sense, the Chan–Lam cross-coupling reaction offers a viable alternative for generating C_aryl_-heteroatom bonds, especially in the arylation of *N*-heterocyclic compounds [11]. This protocol consists of the substitution of an aryl boronic acid with different nucleophiles (especially *N-H*, *O-H*, *S-H*, and *P-H*) in the presence of a copper catalyst under mild reaction conditions (e.g., under air, at room temperature) [12]. Thanks to these characteristics, the *N*-arylation of heterocyclic compounds employing this methodology has become more frequent in the last decade. In particular, the *N*-arylation of pyrroles [13], purines [14] and triazoles [15] is noteworthy and, most recently, the derivatization of a natural product with the *N*-arylation of the quinolizidine alkaloid cytisine has been shown [16].

The *N*-arylation of quinolines, and specifically their 2-oxo derivatives (2-quinolones) **1**, is an emerging area; these *N*-heterocycles are important structural constituents of numerous naturally occurring compounds [17], and represent relevant synthetic scaffolds for the generation of compounds with interesting or useful biological properties, such as as anticancer [18], antimycobacterial [19], insecticidal [20], antibacterial [21], antifungal [22], antiparasitic agents [23], antiviral [24], antimalarial [25], antioxidant [26], and anti-inflammatory properties [27], among others [28,29,30]. Likewise, *N*-arylated 2-quinolones are a common motif in diverse bioactive molecules such as antimicrobials [31], anti-inflammatory drugs [32], sodium channel inhibitors [33], kinase inhibitors [34,35], and HIV inhibitors [36]. Furthermore, the introduction of different substituents through *N*-arylation allows modulation of the electronic and solubility properties of the molecules, which are relevant aspects in the activity of bioactive compounds [35].

The direct *N*-arylation of 2-quinolones through Chan–Lam cross-coupling using phenylboronic acids includes quinolones substituted with -Me, -COOEt, and -OAc groups at positions 4-, 3-, 6, or 7 (Figure 1). In 1999, an article first presented the *N*-arylation of 4,7-dimethyl-2-quinolone (Figure 1, **2**) [37], and later reports describe the *N*-arylation of 3-methoxycarbonyl-2-quinolone **1** using a wide range of substituted phenylboronic acids to form important precursors, such as **3,** to obtain c-Met kinase inhibitors [38]. A synthetic strategy to produce various *N*-aryl-2-quinolones **4** from 2-quinolones in moderate-to-good yields has also been published [36].

In addition to Chan–Lam cross-coupling, a few reports present the direct *N*-arylation of 2-quinolones, including ligand-coupling reactions through organobismuth (V) reagents [39,40] or the use of copper with diverse arylating agents such as aryllead triacetate [41], aryl bromide [42], and substituted diaryliodonium salts [43]. However, the most common methodology for the synthesis of *N*-aryl-quinolones is through intramolecular cyclization, including palladium-catalyzed cross-coupling [44], aromatic nucleophilic substitution (S_N_Ar) [45], and radical pathway reactions such as direct oxidative C-H amidation through visible-light induction [17], and via intramolecular C(sp^2^)-H Knoevenagel products [36]. 

The presence of a substituent at positions 3-, 4-, 6- or 7- in 2-quinolones provides a wide range of possibilities for their functionalization, especially at positions 4- and 3-. In this regard, 3-formyl-2-quinolones have attracted the scientific community’s attention since they are synthetic precursors of great importance for the generation of new compounds with biological properties such as anti-tuberculosis drugs [46], antioxidants [47], anti-cancer [48], antimalarial [49], insecticidal [50] and antibacterial substances [51], and others with possible activity as potentiators of the muscarinic acetylcholine receptor M_4_ related to neurological and psychiatric disorders [52]. They can also be used to produce compounds with luminescent properties [53] and for bioimaging in living cells [54].

Even though the *N*-arylation of 3-formylquinolones has not been widely explored, derivatives of *N*-aryl-3-formylquinolones through CHO functionalization have been subjected to biological testing, such as for antimicrobial [55], antiviral [56], molluscicidal and larvicidal activities [57]; and this allows us to glimpse a future of their potential biological activities. Therefore, due to the synthetic versatility of 3-formylquinolones and the lack of reports on the *N*-arylation of this nucleus, we present here a convenient route for their *N*-arylation through the copper-catalyzed Chan–Lam cross-coupling, using different substituted phenylboronic acids and further elaborating some of the products to produce new more complex chemical entities (Figure 1).

## 2. Results and Discussion

The preparation of 3-formylquinolones was carried out according to the methodology reported by Meth-Cohn et al. [58] (Figure 2). The first step consisted of the acetylation of the commercial anilines **4a**–**e** with acetic anhydride to obtain the respective acetanilides, **5a**–**e**. Cyclization and hydroformylation (in situ) were carried out using the Vilsmeier–Haack reagent, producing the 2-chloro-3-formylquinolines **6a**–**e**. The final step corresponded to acid hydrolysis mediated by 70% acetic acid, which lead to 3-formyl-2-quinolone **7a**–**e** precursors in moderate-to-good yields. 

Our research began with optimization of the synthetic process; we used 3-formylquinolone **7a** as the substrate, 4-methylphenylboronic acid **8a** as the arylating agent, 10 mmol % Cu(OA)_2_ as the catalyst, a 3Å molecular sieve, triethylamine (TEA) as the base, and acetonitrile as the solvent. The entire system was heated at 80 °C for 24 h, but no product formation was observed (Table 1, entry 1). The reason for starting with reagent **8a**, due to the presence of methyl in the *para* (*p*) position, is the generation of an activation in the aromatic ring (electron donor by inductive effect) which can give better performance; this has been evidenced in early reports by the Chan and Lam groups that demonstrated the *N*-arylation of a wide range of *N*-heterocyclic substrates using *p*-methylphenylboronic acid [59]. Furthermore, Janíková et al. reported this acid as a standardization material for Chan–Lam-type *N*-arylation when *N*-heterocyclic systems with carbonyl groups adjacent to an -*NH* group are involved [60]. On the other hand, the low solubility of the compound **7a** in CH_2_Cl_2_ led us to start the standardization with acetonitrile at 80 °C. In turn, the copper salt was not used in stoichiometric amounts since the purpose of this research was to find a protocol where copper is used in catalytic amounts. Lastly, TEA was used as the base, as it is inexpensive, readily available, and according to previous reports, has proved to be exceptionally effective for Chan–Lam coupling between quinolones and phenylboronic acids [61]. However, due to our initial results, we decided to change the base to pyridine, even though at first no arylation product was observed (Table 1, entry 2). Because the low solubility of **7a** in acetonitrile was thought to affect the reaction, the solvent was changed to DMSO, with TEA as the base. Unfortunately, no conversion of the starting material was observed (Table 1, entry 3). For this reason, we tried pyridine again, and obtained the *N*-arylated product **9a** in a modest 15% yield (Table 1, entry 4).

To improve the yield, we changed the solvent to DMF to increase **7a** solubility. TEA was used again, and the yield of the *N*-arylated quinolone **9a** rose to 58% (Table 1, entry 5). Motivated by this result, we began to analyze the base’s effect in this protocol; therefore, different base substances commonly used in Chan–Lam cross-couplings were tried, but the results were similar or even worse than those seen for TEA (Table 1, entries 6–13). Nevertheless, when pyridine was used as the base and DMF as the solvent, the yield rose to 60% (Table 1, entry 14). In addition, using 3Å molecular sieves, we obtained the best yield to the *N*-arylated product **9a**, 64% (Table 1, entry 15). This indicates that, in our system, the presence of water led to a possible competition with the Chan–Lam C-O bond formation [62]. In order to improve the yield of product **9a**, the same reaction conditions mentioned above were explored (Table 1, entry 15), and the reaction time extended up to 48 h; however, the yield of product **9a** did not exceed 64% (Table 1, entry 16). Therefore, it was decided to increase the amount of catalyst to 20 mol%; however, the yield of compound **9a** decreased to 47% (Table 1, test 17). Furthermore, Bipy and TMDA were included as copper ligands (Table 1, entries 18 and 19), but there was no benefit whatsoever. Finally, different Cu(II) salts (Cu (OTf)_2_, CuBr_2_ and CuCl_2_) were used in the hope that they would improve the quality of the proposed protocol (Table 1, entries 20–22); unfortunately, they were less effective. 

It should be noted that 3-formylquinolone systems are tautomerizable heterocycles [63], and because of this, there is the possibility that the Chan–Lam reaction forms *O*-aryl or *N*-aryl bonds, as has been reported with other methodologies [64]. To demonstrate *N*-C_aryl_ bond formation, **9a** was crystallized by slow diffusion at room temperature in a mixture of dichloromethane-ethyl acetate (10:1), and the crystal structure was analyzed using single-crystal X-ray diffraction [CCDC 2205662]. This technique confirmed the structure of the compound, proving that the proposed protocol is selective towards the generation of *N*-aryl-3-formylquinolones (Figure 1). 

Using the optimized conditions, we decided to explore the scope of our protocol employing different *meta-* and *para*-substituted phenylboronic acids. Nineteen derivatives were obtained (**9a**–**s**) on a 1 mmol scale, with yields from 15 to 64%. The *para-* and *meta*-methylphenylboronic acids provided compounds **9a** (64%) and **9c** (30%), the former presenting the highest yield of the synthesized series. Unsubstituted phenylboronic acid led to product **9b** in moderate yield (46%); likewise, *m-tert-buty*lphenylboronic*l* and *p-tert*-butylphenylboronic acids produced **9d** (37%) and **9i** (41%), followed by *m*-methoxyphenylboronic and *p*-methoxyphenylboronic acids that generated **9e** (27%) and **9j** (44%). 

Subsequently, different halogenated phenylboronic acids replaced at the *meta-* and *para*-positions were used. Starting from the *meta*-F, Cl, Br and CF_3_-substituted precursors, the fluoro-derivative **9f** (15%) was obtained with a yield higher than 10%; unfortunately, *meta*-substituted derivatives with Cl, Br, I, CF_3_ and OCF_3_ did not appear in yields greater than 10%. To improve the yields of **9g**–**k**, we opted to analyze the reaction mixtures and found that the main side products were the homocoupled phenylboronic acids. Taking this into account, we decided to add the base and the corresponding phenylboronic acids in portions (0.4 equivalents of each), separated by 90 min over a reaction time of 24 h. Using this methodology for the least satisfactory cases, the yields of the products **9g**, **9h**, and **9j** rose to 45, 41 and 34%, respectively.

On the other hand, the *para*-substituted series provided yields that ranged between 18–35%, the derivative **9q** being the one with the highest transformation (36%) and those with the least, **9s** and **9t** (both 18%), being, now, the analogs **9p**, **9r** and **9u** and reflected yields of 35%, 30% and 20%, respectively. The decrease in yield on changing from *para*- to *meta*-halo derivatives was attributed to the slight increase in acidity and the electronic effect caused by halogens at the *meta* position of the phenylboronic acids, leading to side reactions [65]. 

In the same context, considering both substitution positions, the yields decreased when changing from fluoro to iodo, CF_3_ and OCF_3_, as a result of the electronic effects of these atoms on the acidity of the phenylboronic acid, which leads to byproducts generated by prodeboronation [16,62]. Furthermore, *N*-phenyl derivatives could be obtained with electron-withdrawing groups on the phenyl ring at the *meta*- and *para*-substitutions: methoxycarbonyl gave the *meta*-substitution product **9l** (19%) and the *para*-substituted **9v** (23%), while the formyl group at the *meta* or *p*ara substitution generated **9m** (24%) and **9w** (22%), respectively. Finally, with 2-naphthylboronic acid, **9x** was obtained in 23% yield.

The reaction yields shown in Table 2 apparently depend on the electronic properties and the position of substituents on the aromatic ring of the phenylboronic acid. When there is an electron-donating group in the aromatic system, the reaction yield is higher, in accordance with other Chan–Lam cross-coupling derivatization reports [62,66]. In the same way, the position of the substituent is of great importance. In this case, the *para*-substituted fenilboronic acids give a better conversion towards the *N*-arylated derivative than the *meta*-substituted isomers. This is attributed to the slight increase in acidity presented by the *meta*-substituted phenylboronic acids, since, through the mesomeric and inductive effects exerted by the substituents, the hydroxyborate anion (electrophile) is stabilized in comparison with the *para*-substituted acids [67]. As a consequence, the increased acidity can lead to the formation of undesired products such as phenols, and the oxidation of the aromatic system can lead to quinones [68,69,70] and protodeboronations [71].

It is worth mentioning that *N*-arylation of 3-formylquinolones with heteroaryl boronic acids or with *o*-substituted phenylboronic acids was not achieved (see Appendix A). The *N*-arylated derivatives were not detected and, on the contrary, mixtures of byproducts or decomposition products were obtained. This could be related to the fact that heteroaryl boronic acids are prone to protodeboronations [71], and *o*-substituted phenylboronic acids can undergo intramolecular reactions leading to the formation of hydrogen donor–acceptor bonds that are decisive for the reaction [16,65,72].

To examine the versatility of this methodology, the *N*-arylation of various other 3-formylquinolones (Table 3) was carried out employing phenylboronic acids with a methyl group or a fluorine atom at the *para*-substitution. The series consisted of four substituted 3-formylquinolones bearing a methyl group (**7b**) or a bromine atom (**7c**) at position 6-, and a methoxyl group (**7d**) or a chlorine atom (**7e**) at position 7-. With 6-methyl-3-formylquinolone and *p*-methylphenylboronic acid, **10a** was obtained in a 43% yield, and with *p*-fluorophenylboronic acid, **10b** was produced in a 41% yield. On changing the methyl group to bromine at position 6-, the corresponding *N*-*p*-methylphenyl derivative **10c** was obtained in a higher yield (57%). In contrast, the *N*-*p*-fluorophenyl derivative, **10d**, was acquired in a lower yield (28%) compared with **10b**. According to these results, it is clear that an electron-donating or electron-withdrawing group at position 6- in 3-formylquinolones affects the yield of Chan–Lam cross-couplings. Thus, ongoing from methyl (**7b**) to bromo (**7c**), the *N*-arylation with *p*-methylphenyl boronic acid increases, while it decreases with *para*-fluorophenylboronic acid. These results could be related to the quinolone’s basicity, which drops when electron-withdrawing groups are contained in its structure.

Subsequently, 7-methoxy-3-formylquinolone, **7d**, gave **10e** (32%) and **10f** (26%) with *p*-methoxy- or *p*-fluorophenylboronic acids, respectively. Meanwhile, higher conversions were evidenced from 7-chloro-3-formylquinolone with the aforementioned acids, where the corresponding *N-p*-methylphenyl (**10g**) and *N-p*-fluorophenyl derivatives (**10h**), respectively, were obtained in 50% yield. It is important to highlight that upon exchanging an electron-donating group (at the 6- or 7-position) for a halogen, the yields of *N-p*-methylphenyl derivatives rise. Still, the location of this halogen, together with its electronic nature, could increase or decrease the conversion to the *N-p*-fluorophenyl derivatives. 

With these results, it is clear that varying the electronic nature of the substituent at either the 6- or the 7- position of the 3-formylquinolone affects its basicity [73,74], and therefore has a direct repercussion on the conversion during the Chan–Lam cross-coupling. Notwithstanding, a clear trend is not yet discernible; for that purpose, further experiments should be assessed varying the substituents on the quinolone core.

Thanks to the abovementioned success, we proceeded to demonstrate the chemical reactivity of the coupling products. Therefore, Figure 3 shows different chemical transformations of the aldehyde group at position 3- in **9a**. To establish the versatility of this approach for later derivatizations, we began with a reduction of the aldehyde with NaBH_4_ in MeOH to obtain **11** in 83% yield. Then, the respective nitrile **12** was obtained in 88% yield under fast-heating conditions through a one-pot methodology mediated by “activated” DMSO [75] with hydroxylammonium chloride in DMSO. Additionally, reductive amination of **9a** at room temperature with *p*-methoxybenzylamine in MeOH was carried out to obtain product **13** in 99% yield. Finally, a Claisen–Schmidt type condensation at room temperature between **9a** and 4-methoxyacetophenone, using NaOH (10% *w*/*v*, 0.3 mL) in MeOH as a solvent; this produced chalcone **14** in 83% yield.

## 3. Materials and Methods

### 3.1. General Information

All solvents, including deuterated solvents, were purchased from Merck. Other reagents were from Aldrich, Merck or AK Scientific. Column chromatography was performed on silica gel (Merck, type 60, 0.063–0.2 mm). Melting points were determined on a Reichert Galen III hot plate microscope apparatus and were uncorrected. NMR spectra were recorded on a Bruker Avance 400 MHz spectrometer. All chemical shifts in NMR experiments were reported as ppm downfield from TMS. The following calibrations were used: CDCl_3_ δ = 7.26 and 77.0 ppm for ^1^H NMR and ^13^C NMR, respectively, and DMSO-*d*_6_ δ = 2.50 ppm for ^1^H NMR. Monowave-promoted reactions were performed in a Monowave 50 reactor (Anton Paar, Graz, Austria). HPLC-HR-MS experiments were carried out on an Exactive Plus Orbitrap MS instrument (Thermo Scientific, Waltham, MA, USA). The accurate mass measurements were performed at a resolution of 140,000.

### 3.2. General Procedures and Characterization Data of Compounds

#### 3.2.1. Synthesis of 2-Chloroquinoline-3-carbaldehydes (**6a**–**e**)

These compounds were prepared by following the Meth–Cohn method [58]. DMF (11.6 mL, 150 mmol), in a round-bottom flask was cooled in an ice-water bath to 0.0–2.5 °C and phosphoryl chloride (32.2 mL, 350 mmol) was added dropwise with stirring. To this solution, the corresponding acetanilide (50 mmol) was added and the temperature of the reaction mixture was raised to 80 °C during 20 h. Finally, the mixture was poured into ice-water (300 mL) for 30 min. The precipitate formed was filtered off, washed with cold water and recrystallized from acetonitrile.

##### Synthesis of 2-Oxo-1,2-dihydroquinoline-3-carbaldehyde (3-formyl-2-quinolones **7a**–**e**)

3-Formyl-2-quinolones were synthesized according to the reported procedure [76]. 2-chloroquinoline-3-carbaldehyde (26 mmol) was placed in a 500 mL round-bottom flask followed by 250 mL of 70% AcOH, and the solution was refluxed for 4 h. Finally, Na_2_CO_3_ was added until the mixture reached pH 9. The mixture was filtered and washed with water. The yellow solid was obtained in 80–93% yield. 

##### 2-Oxo-1,2-dihydroquinoline-3-carbaldehyde (**7a**)

Pale yellow solid. ^1^H NMR (200 MHz, DMSO) δ 12.21 (s, 1H), 10.24 (s, 1H), 8.50 (s, 1H), 7.91 (dd, *J* = 7.9, 1.5 Hz, 1H), 7.66 (ddd, *J* = 8.5, 7.1, 1.5 Hz, 1H), 7.36 (d, *J* = 8.3 Hz, 1H), 7.25 (ddd, *J* = 8.2, 7.1, 1.1 Hz, 1H).

##### 6-Methyl-2-oxo-1,2-dihydroquinoline-3-carbaldehyde (**7b**)

Orange solid. ^1^H NMR (200 MHz, DMSO) δ 12.06 (s, 1H), 10.17 (s, 1H), 8.43 (s, 1H), 7.83 (d, *J* = 8.8 Hz, 1H), 6.96–6.77 (m, 2H), 3.86 (s, 3H).

##### 6-Bromo-2-oxo-1,2-dihydroquinoline-3-carbaldehyde (**7c**)

Orange solid. ^1^H NMR (200 MHz, DMSO) δ 12.33 (s, 1H), 10.22 (s, 1H), 8.46 (s, 1H), 8.18 (d, *J* = 2.3 Hz, 1H), 7.78 (dd, *J* = 8.9, 2.3 Hz, 1H), 7.29 (d, *J* = 8.8 Hz, 1H).

##### 7-Chloro-2-oxo-1,2-dihydroquinoline-3-carbaldehyde (**7d**) 

Beige solid. ^1^H NMR (200 MHz, DMSO) δ 12.27 (s, 1H), 10.21 (s, 1H), 8.51 (s, 1H), 7.95 (d, *J* = 8.5 Hz, 1H), 7.46–7.23 (m, 2H).

##### 7-Methoxy-2-oxo-1,2-dihydroquinoline-3-carbaldehyde (**7e**)

Pale yellow solid. ^1^H NMR (200 MHz, DMSO) δ 12.06 (s, 1H), 10.17 (s, 1H), 8.43 (s, 1H), 7.83 (d, *J* = 8.8 Hz, 1H), 6.96–6.77 (m, 2H), 3.86 (s, 3H).

#### 3.2.2. Synthesis of *N*-Aryl-3-formylquinolone Derivatives (**9a**–**x**, **10a**–**h**) 

3-Formyl-2-quinolone (1.0 mmol), substituted-phenyl boronic acid (2 mmol), Cu(OAc)_2_ (18.2 mg, 0.1 mmol), DMF (10 mL), 3 Å molecular sieves (300 mg, 2.5% *w*/*v*) and pyridine (161 µL, 2.0 mmol) were added in a 25 mL round-bottom flask. The final mixture was stirred and heated (open flask) at 80 °C for 24 h. After that, the solvent was removed under a high vacuum and the residue was purified by column chromatography (silica gel, DMC:EtOAc 15:1) to provide the corresponding products in yields going from 15 to 64%.

##### 2-Oxo-1-(p-tolyl)-1,2-dihydroquinoline-3-carbaldehyde (**9a**)

Pale yellow solid; mp: 257–255 °C. ^1^H NMR (400 MHz, CDCl_3_) δ 10.5 (s, 1H), 8.5 (s, 1H), 7.8 (dd, *J* = 7.9, 1.5 Hz, 1H), 7.5–7.4 (m, 3H), 7.3–7.3 (m, 1H), 7.2–7.2 (m, 2H), 6.8 (d, *J* = 8.6 Hz, 1H), 2.5 (s, 3H). ^13^C NMR (101 MHz, CDCl_3_) δ 190.1, 162.1, 143.4, 141.8, 139.4, 134.1, 133.3, 131.3, 131.1, 128.3, 125.8, 123.2, 119.1, 116.4, 21.3. HRMS: m/z [M+H]^+^ Calcd. for C_17_H_13_NO_2_ + H^+^: 264.1024. Found: 264.1040.

##### 2-Oxo-1-phenyl-1,2-dihydroquinoline-3-carbaldehyde (**9b**)

Pale yellow solid; mp: 218–220 °C. ^1^H NMR (400 MHz, CDCl_3_) δ 10.37 (s, 1H), 8.41 (s, 1H), 7.69 (dd, *J* = 7.9, 1.5 Hz, 1H), 7.55 (t, *J* = 7.4 Hz, 2H), 7.48 (t, *J* = 7.4 Hz, 1H), 7.38 (ddd, *J* = 8.7, 7.2, 1.5 Hz, 1H), 7.26–7.14 (m, 3H), 6.62 (d, *J* = 8.6 Hz, 1H). ^13^C NMR (101 MHz, CDCl_3_) δ 189.7, 161.7, 143.0, 141.6, 136.6, 133.2, 131.1, 130.2, 129.1, 128.4, 125.5, 123.0, 118.9, 116.1. HRMS: m/z [M+H]^+^ Calcd. for C_16_H_11_NO_2_ + H^+^: 250.0868. Found: 250.0871.

##### 2-Oxo-1-(m-tolyl)-1,2-dihydroquinoline-3-carbaldehyde (**9c**) 

Pale orange solid; mp: 158–159 °C; ^1^H NMR (400 MHz, CDCl_3_) δ 10.43 (s, 1H), 8.49 (s, 1H), 7.79 (d, *J* = 8.0 Hz, 1H), 7.58–7.44 (m, 2H), 7.37 (d, *J* = 7.4 Hz, 1H), 7.28 (t, *J* = 7.4 Hz, 1H), 7.11 (d, *J* = 10.7 Hz, 2H), 6.74 (t, *J* = 7.6 Hz, 1H), 2.45 (s, 3H). ^13^C NMR (101 MHz, CDCl_3_) δ 189.6, 161.6, 143.0, 141.7, 140.3, 136.5, 133.2, 131.1, 129.9, 128.7, 125.3, 125.2, 123.0, 118.8, 116.1, 21.1. HRMS: m/z [M+H]^+^ Calcd. for C_17_H_13_NO_2_ + H^+^: 264.1024. Found: 264.1032.

##### 1-(3-(*tert*-Butyl)phenyl)-2-oxo-1,2-dihydroquinoline-3-carbaldehyde (**9d**)

Pale beige solid; mp: 118–119 °C; ^1^H NMR (400 MHz, CDCl_3_) δ 10.43 (s, 1H), 8.46 (s, 1H), 7.73 (dd, *J* = 7.9, 1.5 Hz, 1H), 7.59–7.48 (m, 2H), 7.43 (ddd, *J* = 8.6, 7.2, 1.6 Hz, 1H), 7.28–7.18 (m, 2H), 7.08 (dt, *J* = 7.2, 1.8 Hz, 1H), 6.66 (d, *J* = 8.5 Hz, 1H), 1.33 (s, 9H). ^13^C NMR (101 MHz, CDCl_3_) δ 190.0, 162.0, 153.9, 143.5, 141.8, 136.6, 133.4, 131.3, 130.0, 126.4, 125.8, 125.5, 125.4, 123.2, 119.1, 116.4, 35.0, 31.3. HRMS: m/z [M+H]^+^ Calcd. for C_20_H_19_NO_2_ + H^+^: 301.1494. Found: 301.1507.

##### 1-(3-Methoxyphenyl)-2-oxo-1,2-dihydroquinoline-3-carbaldehyde (**9e**)

Pale beige solid; mp: 199–201 °C. ^1^H NMR (400 MHz, CDCl_3_) δ 10.42 (s, 1H), 8.45 (s, 1H), 7.73 (dd, *J* = 7.9, 1.5 Hz, 1H), 7.54–7.40 (m, 2H), 7.27–7.19 (m, 1H), 7.10–7.02 (m, 1H), 6.85 (d, *J* = 7.7 Hz, 1H), 6.79 (t, *J* = 2.2 Hz, 1H), 6.72 (d, *J* = 8.6 Hz, 1H), 3.80 (s, 3H). ^13^C NMR (101 MHz, CDCl_3_) δ 190.0, 161.9, 161.3, 143.2, 141.9, 137.9, 133.4, 131.3, 131.2, 125.8, 123.3, 120.6, 119.1, 116.4, 115.4, 114.0, 55.5. HRMS: m/z [M+H]^+^ Calcd. for C_17_H_13_NO_3_ + H^+^: 280.0973. Found: 280.0981.

##### 1-(3-Fluorophenyl)-2-oxo-1,2-dihydroquinoline-3-carbaldehyde (**9f**)

Brown solid; mp: 216–217 °C. ^1^H NMR (400 MHz, CDCl_3_) δ 10.44 (s, 1H), 8.50 (s, 1H), 7.79 (dd, *J* = 7.8, 1.6 Hz, 1H), 7.62 (td, *J* = 8.1, 6.0 Hz, 1H), 7.50 (ddd, *J* = 8.7, 7.2, 1.5 Hz, 1H), 7.34–7.25 (m, 2H), 7.15–7.11 (m, 1H), 7.07 (dt, *J* = 8.8, 2.2 Hz, 1H), 6.72 (d, *J* = 8.6 Hz, 1H). ^13^C NMR (101 MHz, CDCl_3_) δ 189.64, 164.88, 162.40, 161.72, 142.84, 142.10, 138.14 (d, *J* = 9.6 Hz), 133.58, 131.66 (d, *J* = 9.0 Hz), 124.59 (d, *J* = 3.4 Hz), 116.70 (d, *J* = 20.4 Hz). ^19^F NMR (376 MHz, CDCl_3_) δ-109.65. HRMS: m/z [M+H]^+^ Calcd. for C_16_H_10_FNO_2_ + H^+^: 268.0774. Found: 268.0785.

##### 1-(3-Chlorophenyl)-2-oxo-1,2-dihydroquinoline-3-carbaldehyde (**9g**)

Beige solid; mp: 189–190 °C. ^1^H NMR (400 MHz, CDCl_3_) δ 10.44 (s, 1H), 8.50 (d, *J* = 0.7 Hz, 1H), 7.79 (dd, *J* = 7.8, 1.6 Hz, 1H), 7.61–7.47 (m, 3H), 7.35 (td, *J* = 1.9, 0.6 Hz, 1H), 7.30 (ddd, *J* = 8.0, 7.3, 1.0 Hz, 1H), 7.23 (dt, *J* = 7.0, 2.0 Hz, 1H), 6.71 (dt, *J* = 8.6, 0.8 Hz, 1H). ^13^C NMR (101 MHz, CDCl_3_) δ 189.6, 161.7, 142.8, 142.1, 137.9, 136.0, 133.6, 131.5, 131.4, 129.8, 129.2, 127.1, 125.7, 123.5, 119.1, 116.1. HRMS: m/z [M+H]^+^ Calcd. for C_16_H_10_ClNO_2_ + H^+^: 284.0478. Found: 284.0491.

##### 1-(3-Bromophenyl)-2-oxo-1,2-dihydroquinoline-3-carbaldehyde (**9h**)

Beige solid; mp: 199–200 °C. ^1^H NMR (400 MHz, CDCl_3_) δ 10.44 (s, 1H), 8.50 (s, 1H), 7.79 (dd, *J* = 7.8, 1.5 Hz, 1H), 7.71 (ddd, *J* = 8.1, 1.9, 1.0 Hz, 1H), 7.56–7.46 (m, 3H), 7.34–7.24 (m, 2H), 6.71 (d, *J* = 8.6 Hz, 1H).. ^13^C NMR (101 MHz, CDCl_3_) δ 189.6, 161.8, 142.8, 142.1, 138.0, 133.6, 132.7, 132.0, 131.6, 131.5, 127.6, 125.7, 123.7, 123.6, 119.1, 116.1. HRMS: m/z [M+H]^+^ Calcd. for C_16_H_10_BrNO_2_ + H^+^: 327.9973. Found: 327.9984.

##### 1-(3-Iodophenyl)-2-oxo-1,2-dihydroquinoline-3-carbaldehyde (**9i**)

Pale green solid; mp: 214–215 °C. ^1^H NMR (400 MHz, CDCl_3_) δ 10.44 (s, 1H), 8.49 (d, *J* = 0.7 Hz, 1H), 7.91 (dt, *J* = 7.8, 1.4 Hz, 1H), 7.78 (dd, *J* = 7.8, 1.5 Hz, 1H), 7.68 (t, *J* = 1.8 Hz, 1H), 7.51 (ddd, *J* = 8.7, 7.2, 1.5 Hz, 1H), 7.38 (t, *J* = 7.9 Hz, 1H), 7.33–7.27 (m, 2H), 6.71 (dd, *J* = 8.6, 0.9 Hz, 1H). ^13^C NMR (101 MHz, CDCl_3_) δ 189.6, 161.7, 142.8, 142.1, 138.6, 137.9, 137.6, 133.6, 131.7, 131.5, 128.2, 125.7, 123.5, 119.1, 116.1, 94.9. HRMS: m/z [M+H]^+^ Calcd. for C_16_H_10_INO_2_ + H^+^: 375.9834. Found: 375.9847.

##### 2-Oxo-1-(3-(trifluoromethyl)phenyl)-1,2-dihydroquinoline-3-carbaldehyde (**9j**)

Orange solid; mp: 170–172 °C. ^1^H NMR (400 MHz, CDCl_3_) δ 10.44 (s, 1H), 8.52 (s, 1H), 7.88–7.73 (m, 3H), 7.62 (d, *J* = 1.9 Hz, 1H), 7.58–7.47 (m, 2H), 7.31 (ddd, *J* = 8.1, 7.3, 1.0 Hz, 1H), 6.65 (d, *J* = 8.5 Hz, 1H). ^13^C NMR (101 MHz, CDCl_3_) δ 189.5, 161.8, 142.8, 142.3, 137.4, 133.7, 133.1 (d, *J* = 33.2 Hz), 132.5, 131.7, 131.1, 126.4 (d, *J* = 3.7 Hz), 126.0 (d, *J* = 3.9 Hz), 125.7, 123.7, 119.2. ^19^F NMR (376 MHz, CDCl_3_) δ-62.6. HRMS: m/z [M+H]^+^ Calcd. for C_17_H_10_F_3_NO_2_ + H^+^: 318.0742. Found: 318.0758.

##### 2-Oxo-1-(3-(trifluoromethoxy)phenyl)-1,2-dihydroquinoline-3-carbaldehyde (**9k**)

Orange solid; mp: 142–144 °C. ^1^H NMR (400 MHz, CDCl_3_) δ 10.4 (s, 1H), 8.5 (s, 1H), 7.8 (d, *J* = 7.8 Hz, 1H), 7.7 (t, *J* = 8.2 Hz, 1H), 7.5 (d, *J* = 8.0 Hz, 1H), 7.4 (d, *J* = 8.4 Hz, 1H), 7.3 (dt, *J* = 9.0, 6.5 Hz, 3H), 6.7 (d, *J* = 8.6 Hz, 1H). ^13^C NMR (101 MHz, CDCl_3_) δ 189.5, 161.7, 150.5 (d, *J* = 2.1 Hz), 142.8, 142.2, 138.1, 133.6, 131.6 (d, *J* = 4.0 Hz), 127.3, 125.7, 123.6, 121.7, 119.2, 115.9. ^19^F NMR (376 MHz, CDCl_3_) δ-57.9. HRMS: m/z [M+H]^+^ Calcd. for C_17_H_10_F_3_NO_3_ + H^+^: 334.0691. Found: 334.0703.

##### Methyl 3-(3-formyl-2-oxoquinolin-1(2H)-yl)benzoate (**9l**)

Pale green solid; mp: 224–225 °C. ^1^H NMR (400 MHz, CDCl_3_) δ 10.4 (s, 1H), 8.4 (s, 1H), 8.2 (d, *J* = 7.9 Hz, 1H), 7.9 (t, *J* = 1.9 Hz, 1H), 7.7–7.7 (m, 1H), 7.6 (t, *J* = 7.9 Hz, 1H), 7.5–7.3 (m, 2H), 7.2 (t, *J* = 6.8 Hz, 1H), 6.6 (d, *J* = 8.6 Hz, 1H), 3.8 (s, 3H). ^13^C NMR (101 MHz, CDCl_3_) δ 189.6, 165.8, 161.8, 142.9, 142.1, 137.0, 133.6, 133.3, 132.7, 131.6, 130.6, 130.5, 130.1, 125.7, 123.5, 119.2, 116.1, 52.4. HRMS: m/z [M+H]^+^ Calcd. for C_18_H_13_NO_4_ + H^+^: 308.0923. Found: 308.0934.

##### 1-(3-Formylphenyl)-2-oxo-1,2-dihydroquinoline-3-carbaldehyde (**9m**)

Pale yellow solid; mp: 158–160 °C. ^1^H NMR (400 MHz, CDCl_3_) δ 10.4 (s, 1H), 10.1 (s, 1H), 8.5 (s, 1H), 8.1–8.1 (m, 1H), 7.9–7.8 (m, 3H), 7.6 (ddd, *J* = 7.9, 2.1, 1.2 Hz, 1H), 7.5 (ddd, *J* = 8.7, 7.2, 1.6 Hz, 1H), 7.4–7.3 (m, 1H), 6.7 (d, *J* = 8.6 Hz, 1H). ^13^C NMR (101 MHz, CDCl_3_) δ 190.8, 189.5, 161.8, 142.7, 142.3, 138.5, 137.8, 134.8, 133.7, 131.7, 131.2, 130.5, 129.9, 125.6, 123.7, 119.2, 115.9. HRMS: m/z [M+H]^+^ Calcd. for C_17_H_11_NO_3_ + H^+^: 278.0817. Found: 278.0829.

##### 1-(4-(*tert*-Butyl)phenyl)-2-oxo-1,2-dihydroquinoline-3-carbaldehyde (**9n**)

Pale yellow solid; mp: 209–210 °C. ^1^H NMR (400 MHz, CDCl_3_) δ 10.5 (s, 1H), 8.5 (s, 1H), 7.8 (dd, *J* = 7.8, 1.5 Hz, 1H), 7.7–7.6 (m, 2H), 7.5–7.4 (m, 1H), 7.3–7.3 (m, 1H), 7.2–7.2 (m, 2H), 6.8 (d, *J* = 8.3 Hz, 1H), 1.4 (s, 9H). ^13^C NMR (101 MHz, CDCl_3_) δ 190.1, 162.1, 152.4, 143.4, 141.7, 134.0, 133.3, 131.3, 127.9, 127.4, 125.7, 123.2, 119.1, 116.5, 34.9, 31.4. HRMS: m/z [M+H]^+^ Calcd. for C_20_H_19_NO_2_ + H^+^: 306.1494. Found: 306.1510.

##### 1-(4-Methoxyphenyl)-2-oxo-1,2-dihydroquinoline-3-carbaldehyde (**9o**)

Yellow-greenish solid; mp: 250–251 °C. ^1^H NMR (400 MHz, CDCl_3_) δ 10.46 (s, 1H), 8.48 (s, 1H), 7.76 (dd, *J* = 7.9, 1.5 Hz, 1H), 7.47 (ddd, *J* = 8.7, 7.2, 1.6 Hz, 1H), 7.30–7.17 (m, 3H), 7.15–7.11 (m, 2H), 6.78 (d, *J* = 8.6 Hz, 1H), 3.90 (s, 3H). ^13^C NMR (101 MHz, CDCl_3_) δ 190.1, 162.2, 160.0, 143.6, 141.7, 133.4, 131.3, 129.6, 129.2, 125.8, 123.2, 119.1, 116.4, 115.6, 55.6. HRMS: m/z [M+H]^+^ Calcd. for C_17_H_13_NO_3_ + H^+^: 280.0973. Found: 280.0984.

##### 1-(4-Fluorophenyl)-2-oxo-1,2-dihydroquinoline-3-carbaldehyde (**9p**)

Pale green solid; mp: 244–245 °C. ^1^H NMR (400 MHz, CDCl_3_) δ 10.4 (s, 1H), 8.4 (s, 1H), 7.7 (dd, *J* = 7.8, 1.5 Hz, 1H), 7.4 (ddd, *J* = 8.7, 7.2, 1.6 Hz, 1H), 7.3–7.2 (m, 5H), 6.6 (d, *J* = 8.6 Hz, 1H). ^13^C NMR (101 MHz, CDCl_3_) δ 189.7, 164.1, 162.0, 161.6, 143.2, 142.0, 133.5, 132.6 (d, *J* = 3.5 Hz), 131.5, 130.5 (d, *J* = 8.7 Hz), 125.7, 123.4, 119.2, 117.5 (d, *J* = 23.0 Hz), 116.1. ^19^F NMR (376 MHz, CDCl_3_) δ -111.4. HRMS: m/z [M+H]^+^ Calcd. for C_16_H_10_FNO_2_ + H^+^: 268.0774. Found: 268.0791.

##### 1-(4-Chlorophenyl)-2-oxo-1,2-dihydroquinoline-3-carbaldehyde (**9q**)

Pale green solid; mp: 235–237 °C. ^1^H NMR (400 MHz, CDCl_3_) δ 10.4 (s, 1H), 8.5 (s, 1H), 7.8 (dd, *J* = 7.8, 1.5 Hz, 1H), 7.6–7.6 (m, 2H), 7.5 (ddd, *J* = 8.7, 7.2, 1.5 Hz, 1H), 7.3–7.2 (m, 3H), 6.7 (d, *J* = 8.5 Hz, 1H). ^13^C NMR (101 MHz, CDCl_3_) δ 189.6, 161.8, 143.0, 142.1, 135.4, 135.3, 133.6, 131.6, 130.7, 130.1, 125.7, 123.5, 119.2, 116.1. HRMS: m/z [M+H]^+^ Calcd. for C_16_H_10_ClNO_2_ + H^+^: 284.0478. Found: 284.0491.

##### 1-(4-Bromophenyl)-2-oxo-1,2-dihydroquinoline-3-carbaldehyde (**9r**)

Pale green solid; mp: 250–253 °C. ^1^H NMR (400 MHz, CDCl_3_) δ 10.4 (s, 1H), 8.5 (d, *J* = 0.7 Hz, 1H), 7.8–7.7 (m, 3H), 7.5 (ddd, *J* = 8.7, 7.2, 1.5 Hz, 1H), 7.3–7.3 (m, 1H), 7.2–7.2 (m, 2H), 6.7 (dd, *J* = 8.6, 0.9 Hz, 1H). ^13^C NMR (101 MHz, CDCl_3_) δ 189.7, 161.8, 142.9, 142.1, 135.8, 133.7, 133.6, 131.6, 130.5, 125.7, 123.5, 123.5, 119.2, 116.1. HRMS: m/z [M+H]^+^ Calcd. for C_16_H_10_BrNO_2_ + H^+^: 327.9997. Found: 328.0011.

##### 1-(4-Iodophenyl)-2-oxo-1,2-dihydroquinoline-3-carbaldehyde (**9s**)

Pale green solid; mp: 264–265 °C. ^1^H NMR (400 MHz, CDCl_3_) δ 10.4 (s, 1H), 8.5 (s, 1H), 8.0–7.9 (m, 2H), 7.8 (dd, *J* = 7.8, 1.5 Hz, 1H), 7.5–7.4 (m, 1H), 7.3–7.2 (m, 1H), 7.1–7.0 (m, 2H), 6.7 (d, *J* = 8.6 Hz, 1H). ^13^C NMR (101 MHz, CDCl_3_) δ 189.7, 161.7, 142.8, 142.0, 139.7, 136.5, 133.5, 131.5, 130.6, 125.7, 123.5, 119.1, 116.1, 95.1. HRMS: m/z [M+H]^+^ Calcd. for C_16_H_10_INO_2_ + H^+^: 376.9834. Found: 376.9844.

##### 2-Oxo-1-(4-(trifluoromethyl)phenyl)-1,2-dihydroquinoline-3-carbaldehyde (**9t**)

Pale orange solid; mp: 258–260 °C. ^1^H NMR (400 MHz, DMSO) δ 10.3 (d, *J* = 2.2 Hz, 1H), 8.7 (d, *J* = 16.8 Hz, 1H), 8.2–7.9 (m, 3H), 7.7–7.5 (m, 3H), 7.4–7.3 (m, 1H), 6.6 (d, *J* = 8.5 Hz, 1H). ^13^C NMR (101 MHz, CDCl_3_) δ 188.8, 161.0, 142.2 (d, *J* = 9.5 Hz), 140.0, 133.4, 130.6 (d, *J* = 32.5 Hz), 129.5, 127.1 (q, *J* = 3.8 Hz), 124.9 (d, *J* = 30.1 Hz), 123.3, 118.7, 115.4. ^19^F NMR (376 MHz, CDCl_3_) δ -62.7. HRMS: m/z [M+H]^+^ Calcd. for C_17_H_10_F_3_NO_2_ + H^+^: 318.0742. Found: 3108.07452.

##### 2-Oxo-1-(4-(trifluoromethoxy)phenyl)-1,2-dihydroquinoline-3-carbaldehyde (**9u**)

Pale brown solid; mp: 219–221 °C. ^1^H NMR (400 MHz, CDCl_3_) δ 10.3 (s, 1H), 8.3 (s, 1H), 7.6 (dd, *J* = 7.8, 1.5 Hz, 1H), 7.4–7.3 (m, 3H), 7.2–7.2 (m, 2H), 7.2–7.1 (m, 1H), 6.6 (d, *J* = 8.6 Hz, 1H). ^13^C NMR (101 MHz, CDCl_3_) δ 189.6, 161.9, 149.6 (d, *J* = 2.0 Hz), 142.9, 142.1, 135.1, 133.6, 131.6, 130.4, 125.7, 123.6, 122.7, 119.2, 116.0. ^19^F NMR (376 MHz, CDCl_3_) δ -57.8. HRMS: m/z [M+H]^+^ Calcd. for C_17_H_10_F_3_NO_3_ + H^+^: 334.0691. Found: 334.0705.

##### Methyl 4-(3-Formyl-2-oxoquinolin-1(2H)-yl)benzoate (**9v**)

Greenish-yellow solid; mp: 217–219 °C. ^1^H NMR (400 MHz, CDCl_3_) δ 10.42 (s, OH), 8.50 (s, 1H), 8.31 (d, *J* = 7.1 Hz, 1H), 7.80 (d, *J* = 7.9 Hz, 1H), 7.53–7.41 (m, 1H), 7.45–7.38 (m, 2H), 7.34–7.26 (m, 1H), 6.68 (d, *J* = 8.6 Hz, 1H), 3.99 (s, 1H). ^13^C NMR (101 MHz, CDCl_3_) δ 189.5, 166.0, 161.6, 142.7, 142.2, 140.9, 133.6, 131.7, 131.6, 131.2, 129.0, 125.6, 123.5, 119.1, 116.0, 52.5. HRMS: m/z [M+H]^+^ Calcd. for C_18_H_13_NO_4_ + H^+^: 308.0923. Found: 308.0933.

##### 1-(4-Formylphenyl)-2-oxo-1,2-dihydroquinoline-3-carbaldehyde (**9w**)

Pale green solid; mp: 233–235 °C. ^1^H NMR (400 MHz, CDCl_3_) δ 10.43 (s, 1H), 10.16 (s, 1H), 8.52 (s, 1H), 8.22–8.13 (m, 2H), 7.82 (dd, *J* = 7.9, 1.5 Hz, 1H), 7.57–7.46 (m, 3H), 7.36–7.28 (m, 1H), 6.68 (dd, *J* = 8.6, 1.0 Hz, 1H). ^13^C NMR (101 MHz, CDCl_3_) δ 191.0, 189.4, 161.6, 142.5, 142.3, 142.2, 136.9, 133.7, 131.7, 131.6, 129.8, 125.7, 123.7, 119.2, 115.9. HRMS: m/z [M+H]^+^ Calcd. for C_17_H_11_NO_3_ + H^+^: 278.0817. Found: 278.0834.

##### 1-(Naphthalen-2-yl)-2-oxo-1,2-dihydroquinoline-3-carbaldehyde (**9x**)

Pale green solid; mp: 256–257 °C. ^1^H NMR (400 MHz, CDCl_3_) δ 10.48 (s, 1H), 8.53 (s, 1H), 8.10 (d, *J* = 8.6 Hz, 1H), 7.97 (d, *J* = 7.9 Hz, 1H), 7.89 (d, *J* = 7.7 Hz, 1H), 7.84 (d, *J* = 2.1 Hz, 1H), 7.79 (dd, *J* = 7.9, 1.6 Hz, 1H), 7.69–7.52 (m, 2H), 7.43 (ddd, *J* = 8.7, 7.2, 1.6 Hz, 1H), 7.37 (dd, *J* = 8.6, 2.1 Hz, 1H), 7.27 (t, *J* = 6.1 Hz, 1H), 6.74 (d, *J* = 8.6 Hz, 1H). ^13^C NMR (101 MHz, CDCl_3_) δ 190.0, 162.2, 143.4, 141.9, 134.2, 134.0, 133.4, 133.4, 131.4, 130.6, 128.2, 128.0, 127.8, 127.3, 127.0, 125.9, 125.8, 123.3, 119.2, 116.5. HRMS: m/z [M+H]^+^ Calcd. for C_20_H_13_NO_2_ + H^+^: 300.1024. Found: 300.1035.

##### 6-Methyl-2-oxo-1-(p-tolyl)-1,2-dihydroquinoline-3-carbaldehyde (**10a**)

Greenish-yellow solid; mp: 268–269 °C. ^1^H NMR (400 MHz, CDCl_3_) δ 10.37 (s, 1H), 8.34 (s, 1H), 7.48–7.43 (m, 1H), 7.34 (d, *J* = 8.0 Hz, 2H), 7.23–7.16 (m, 1H), 7.13–7.05 (m, 2H), 6.57 (d, *J* = 8.7 Hz, 1H), 2.40 (s, 3H), 2.33 (s, 3H). ^13^C NMR (101 MHz, CDCl_3_) δ 190.2, 162.0, 141.6, 141.5, 139.3, 134.8, 134.3, 132.9, 131.0, 130.8, 128.3, 125.7, 119.1, 116.3, 21.3, 20.5. HRMS: m/z [M+H]^+^ Calcd. for C_18_H_15_NO_2_ + H^+^: 278.1181. Found: 278.1199.

##### 1-(4-Fluorophenyl)-6-methyl-2-oxo-1,2-dihydroquinoline-3-carbaldehyde (**10b**)

Green solid; mp: 277–278 °C. ^1^H NMR (400 MHz, CDCl_3_) δ 10.44 (s, 1H), 8.43 (s, 1H), 7.58–7.53 (m, 1H), 7.35–7.27 (m, 5H), 6.61 (d, *J* = 8.7 Hz, 1H), 2.42 (s, 3H). ^13^C NMR (101 MHz, CDCl_3_) δ 189.9, 164.0, 162.0, 161.5, 141.8, 141.3, 135.0, 133.2, 132.7 (d, *J* = 3.3 Hz), 131.0, 130.5 (d, *J* = 8.8 Hz), 125.6, 119.1, 117.4 (d, *J* = 23.0 Hz), 116.0, 20.5. ^19^F NMR (376 MHz, CDCl_3_) δ-111.6. HRMS: m/z [M+H]^+^ Calcd. for C_17_H_12_FNO_2_ + H^+^: 282.0930. Found: 282.0941.

##### 6-Bromo-2-oxo-1-(p-tolyl)-1,2-dihydroquinoline-3-carbaldehyde (**10c**)

Brown solid; mp: 265–266 °C. ^1^H NMR (400 MHz, CDCl_3_) δ 10.34 (s, 1H), 8.28 (s, 1H), 7.79 (d, *J* = 2.2 Hz, 1H), 7.42 (dd, *J* = 9.1, 2.3 Hz, 1H), 7.34 (d, *J* = 7.6 Hz, 2H), 7.07 (d, *J* = 8.2 Hz, 2H), 6.56 (d, *J* = 9.0 Hz, 1H), 2.39 (s, 3H). ^13^C NMR (101 MHz, CDCl_3_) δ 189.6, 161.6, 142.2, 140.3, 139.7, 135.9, 133.7, 133.0, 131.2, 128.1, 126.5, 120.4, 118.1, 115.9, 21.3. HRMS: m/z [M+H]^+^ Calcd. for C_17_H_12_BrNO_2_ + H^+^: 342.0129. Found: 342.0144.

##### 6-Bromo-1-(4-fluorophenyl)-2-oxo-1,2-dihydroquinoline-3-carbaldehyde (**10d**)

Orange solid; mp: 271–272 °C. ^1^H NMR (400 MHz, CDCl_3_) δ 10.42 (s, 1H), 8.38 (s, 1H), 7.90 (d, *J* = 2.3 Hz, 1H), 7.55 (dd, *J* = 9.1, 2.2 Hz, 1H), 7.34–7.24 (m, 4H), 6.61 (d, *J* = 9.0 Hz, 1H). ^13^C NMR (101 MHz, CDCl_3_) δ 189.33, 164.17, 161.69, 161.53, 142.00, 140.56, 136.11, 133.23, 132.18 (d, *J* = 3.5 Hz), 130.41 (d, *J* = 8.8 Hz), 126.53, 120.49, 117.68 (d, *J* = 23.1 Hz), 116.18. ^19^F NMR (376 MHz, CDCl_3_) δ-110.9. HRMS: m/z [M+H]^+^ Calcd. for C_16_H_9_BrFNO_2_ + H^+^: 345.9879. Found: 345.9891.

##### 7-Methoxy-2-oxo-1-(p-tolyl)-1,2-dihydroquinoline-3-carbaldehyde (**10e**)

Pale beige solid; mp: 187–188 °C. ^1^H NMR (400 MHz, CDCl_3_) δ 10.37 (s, 1H), 8.39 (s, 1H), 7.64 (d, *J* = 8.7 Hz, 1H), 7.43–7.37 (m, 2H), 7.20–7.12 (m, 2H), 6.83 (dd, *J* = 8.8, 2.4 Hz, 1H), 6.13 (d, *J* = 2.3 Hz, 1H), 3.69 (s, 3H), 2.46 (s, 3H). ^13^C NMR (101 MHz, CDCl_3_) δ 189.9, 164.1, 162.6, 145.7, 141.6, 139.3, 134.3, 133.1, 131.1, 128.2, 122.9, 113.5, 111.8, 100.2, 55.6, 21.3. HRMS: m/z [M+H]^+^ Calcd. for C_18_H_15_NO_3_ + H^+^: 294.1052. Found: 294.1066.

##### 1-(4-Fluorophenyl)-7-methoxy-2-oxo-1,2-dihydroquinoline-3-carbaldehyde (**10f**)

Pale brown solid; mp: 213–214 °C. ^1^H NMR (400 MHz, CDCl_3_) δ 10.29 (s, 1H), 8.35 (s, 1H), 7.62 (d, *J* = 8.8 Hz, 1H), 7.24 (dd, *J* = 6.5, 2.0 Hz, 4H), 6.80 (dd, *J* = 8.8, 2.4 Hz, 1H), 6.03 (d, *J* = 2.3 Hz, 1H), 3.66 (s, 3H). ^13^C NMR (101 MHz, CDCl_3_) δ 189.6, 164.3, 164.0, 162.5, 161.5, 145.4, 141.9, 133.4, 132.8 (d, *J* = 3.4 Hz), 130.5 (d, *J* = 8.8 Hz), 122.8, 117.5 (d, *J* = 22.9 Hz), 113.5, 111.8, 100.1, 55.6. ^19^F NMR (376 MHz, CDCl_3_) δ -111.4. HRMS: m/z [M+H]^+^ Calcd. for C_17_H_12_FNO_3_ + H^+^: 298.0879. Found: 298.0888.

##### 7-Chloro-2-oxo-1-(p-tolyl)-1,2-dihydroquinoline-3-carbaldehyde (**10g**)

Pale beige solid; mp: 251–253 °C. ^1^H NMR (400 MHz, CDCl_3_) δ 10.41 (s, 1H), 8.42 (s, 1H), 7.68 (d, *J* = 8.3 Hz, 1H), 7.44 (d, *J* = 8.0 Hz, 2H), 7.25–7.18 (m, 1H), 7.21–7.12 (m, 2H), 6.73 (d, *J* = 1.8 Hz, 1H), 2.49 (s, 3H). ^13^C NMR (101 MHz, CDCl_3_) δ 189.6, 161.8, 144.0, 141.0, 140.0, 139.8, 133.5, 132.4, 131.3, 128.1, 125.6, 123.9, 117.5, 116.2, 21.4. HRMS: m/z [M+H]^+^ Calcd. for C_17_H_12_ClNO_2_ + H^+^: 298.0635. Found: 298.0649.

##### 7-Chloro-1-(4-fluorophenyl)-2-oxo-1,2-dihydroquinoline-3-carbaldehyde (**10h**)

Greenish-orange solid; mp: 268–269 °C. ^1^H NMR (400 MHz, CDCl_3_) δ 10.40 (s, 1H), 8.44 (s, 1H), 7.71 (d, *J* = 8.4 Hz, 1H), 7.38–7.24 (m, 5H), 6.70 (d, *J* = 1.9 Hz, 1H). ^13^C NMR (101 MHz, CDCl_3_) δ 189.3, 164.2, 161.7, 161.7, 143.8, 141.2, 140.2, 132.6, 132.0 (d, *J* = 3.4 Hz), 130.4 (d, *J* = 8.8 Hz), 125.6, 124.1, 117.8 (d, *J* = 23.0 Hz), 116.0. ^19^F NMR (376 MHz, CDCl_3_) δ-110.7. HRMS: m/z [M+H]^+^ Calcd. for C_16_H_9_ClFNO_2_ + H^+^: 302.0384. Found: 302.0399.

#### 3.2.3. Synthesis of 3-(Hydroxymethyl)-1-(p-tolyl)quinolin-2(1H)-one (**11**)

Compound **11** was synthetized according to the reported procedure [77], **9a** (0.3 mmol) was added to a vial containing MeOH (4.5 mL), and NaBH_4_ (0.9 mmol) was added in small portions. After complete disappearance of the starting material (monitored by TLC), the solution was stirred for another 10 min. Finally, the solvent was removed in a vacuum, and the solid washed with cold water (5 mL) and filtered, giving a brown solid (83% yield).

Brown solid; mp: 169–171 °C. ^1^H NMR (400 MHz, CDCl_3_) δ 7.78 (s, 1H), 7.58 (d, *J* = 7.7 Hz, 1H), 7.42–7.28 (m, 3H), 7.24–7.11 (m, 3H), 6.72 (d, *J* = 8.5 Hz, 1H), 4.67 (s, 2H), 3.74 (s, 1H), 2.45 (s, 3H). ^13^C NMR (101 MHz, CDCl_3_) δ 162.6, 140.4, 139.1, 135.7, 134.6, 132.0, 130.9, 129.8, 128.4, 128.4, 122.7, 120.2, 116.0, 62.5, 21.3. HRMS: m/z [M+H]^+^ Calcd. for C_17_H_15_NO_2_ + H^+^: 266.1181. Found: 266.1193.

#### 3.2.4. Synthesis of 2-Oxo-1-(p-tolyl)-1,2-dihydroquinoline-3-carbonitrile (**12**)

A G10 reaction vial was charged with **9a** (0.3 mmol), hydroxylammonium chloride (0.36 mmol) and DMSO (2.5 mL). The vial was capped, placed in the Mono-Wave reactor and heated at 110 °C for 1 h and 30 min. Then, water (5 mL) was added, the precipitate was filtered and washed with abundant water. The solid was purified by column chromatography to give a beige solid (88% yield).

Beige solid; mp: 228–229 °C. ^1^H NMR (400 MHz, CDCl_3_) δ 8.33 (s, 1H), 7.68 (dd, *J* = 7.8, 1.5 Hz, 1H), 7.48 (ddd, *J* = 8.7, 7.2, 1.6 Hz, 1H), 7.40 (d, *J* = 7.8 Hz, 2H), 7.33–7.24 (m, 1H), 7.14 (d, *J* = 8.3 Hz, 2H), 6.75 (d, *J* = 8.6 Hz, 1H), 2.46 (s, 3H). ^13^C NMR (101 MHz, CDCl_3_) δ 158.6, 148.2, 142.5, 139.7, 133.7, 133.7, 131.1, 129.8, 128.1, 123.4, 118.6, 116.6, 115.1, 107.6, 21.3. HRMS: m/z [M+H]^+^ Calcd. for C_17_H_12_N_2_O+H^+^: 261.1028. Found: 261.1041.

#### 3.2.5. Synthesis of 3-(((4-Methoxybenzyl)amino)methyl)-1-(p-tolyl)quinolin-2(1H)-one (**13**)

Compound **13** was synthetized according to the reported procedure [78]. A mixture of quinoline-3-carbaldehyde **9a** (0.3 mmol) and 4-methoxybenzylamine (0.32 mmol) in MeOH (1.2 mL) was stirred for 4 h at room temperature. Subsequently, solid NaBH_4_ was added (0.48 mmol) portion-wise and the stirring was continued at ambient temperature for 20 min. After the reaction was complete, the mixture was poured into water (30 mL) and extracted with EtOAc (50 mL × 2). The combined organic layer-phases were dried over anhydrous Na_2_SO_4_, (5 mL × 3 times), filtered and concentrated in vacuo. After removal the solvent, the crude was purified by column chromatography (EtOAc:MeOH 4:1) to give a brown solid (99% yield). 

Waxy brown solid. ^1^H NMR (400 MHz, CDCl_3_) δ 7.80 (s, 1H), 7.58 (dd, *J* = 7.8, 1.5 Hz, 1H), 7.39 (d, *J* = 8.5 Hz, 2H), 7.34–7.24 (m, 3H), 7.23–7.10 (m, 3H), 6.89–6.83 (m, 2H), 6.70 (dd, *J* = 8.5, 1.0 Hz, 1H), 3.83 (d, *J* = 6.1 Hz, 4H), 3.78 (s, 3H), 2.46 (s, 3H). ^13^C NMR (101 MHz, CDCl_3_) δ 162.5, 158.7, 140.5, 138.8, 136.9, 135.0, 131.9, 131.3, 130.8, 129.6, 129.5, 128.4, 128.1, 122.4, 120.3, 115.9, 113.9, 55.3, 52.8, 49.5, 21.3. HRMS: m/z [M+H]^+^ Calcd. for C_25_H_24_N_2_O_2_+H^+^: 385.1916. Found: 385.1922.

#### 3.2.6. Synthesis of (E)-3-(3-(4-Methoxyphenyl)-3-oxoprop-1-en-1-yl)-1-(p-tolyl)quinolin-2(1H)-one (**14**)

Compound **14** was synthetized according to the reported procedure [79,80,81]. **9a** (0.3 mmol), and 4-methoxyacetophenone (0.3 mmol) were added to a vial containing MeOH (1.5 mL), the solution was stirred and NaOH (sol. 10% *w*/*v*, 0.3 mL) was added dropwise. The final solution was stirred for 24 h, and the solid was filtered, washed with cold MeOH and dried in a vacuum to give a pale green solid (83% yield).

Pale green solid; mp 217–219 °C. ^1^H NMR (400 MHz, CDCl_3_) δ 8.64 (d, *J* = 15.3 Hz, 1H), 8.32–8.06 (m, 2H), 8.03 (s, 1H), 7.74 (d, *J* = 15.3 Hz, 1H), 7.64 (dd, *J* = 7.9, 1.5 Hz, 1H), 7.43 (d, *J* = 8.0 Hz, 2H), 7.39–7.30 (m, 1H), 7.24–7.14 (m, 3H), 6.92–6.87 (m, 2H), 6.69 (d, *J* = 8.5 Hz, 1H), 3.83 (s, 3H), 2.47 (s, 3H). ^13^C NMR (101 MHz, CDCl_3_) δ 189.2, 163.5, 161.3, 143.1, 140.9, 139.1, 139.1, 135.0, 131.4, 131.1, 131.1, 131.1, 129.1, 128.4, 126.5, 126.2, 122.7, 120.1, 116.0, 113.7, 55.5, 21.3. HRMS: m/z [M+H]^+^ Calcd. for C_26_H_21_NO_3_ + H^+^: 396.1599. Found: 396.1611.

## 4. Conclusions

In conclusion, a versatile and selective copper-catalyzed Chan–Lam protocol for the *N*-arylation of several 3-formylquinolones has been developed. The protocol includes catalytic amounts of copper(II) with inexpensive phenylboronic acids in open flasks. Electron-donating or electron-withdrawing groups and halogens at the *para*- and *meta*-positions of the phenylboronic acids were tolerated. The protocol was suitable for *N*-arylation of four 3-formylquinolones substituted at positions 6- and 7-. Finally, diversification of *N-p*-methylphenyl derivative **9a** through aldehyde modification at position 3- of the quinolone core was possible, demonstrating the potential of the new compound for synthetic transformations. Therefore, this protocol is a viable synthetic tool to obtain *N*-arylated derivatives that can be used to generate compounds with biological, luminescent, and catalytic properties, among others.

## Data Availability

Not applicable.

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
