# Peer review of "N-Arylation of 3-Formylquinolin-2(1H)-ones Using Copper(II)-Catalyzed Chan–Lam Coupling"

_molecules, 2022, doi:10.3390/molecules27238345_

Round 1

Reviewer 1 Report

The manuscript “N-Arylation of 3-formylquinolin-2(1H)-ones by Copper(II)-cata-2 lyzed Cham-Lam coupling” reports the optimization of reaction conditions of Chan-Lam C-N coupling between boronic acids and 3-formylquinolin-2(1H)-ones, the scope of this reaction and examples of synthetic applications of products. Similar reactions were realized previously for other quinolin-2(1H)-ones, the manuscript represents the extension of these studies. In general, the manuscript is well-written and experimental procedures are reported in good detail. In my opinion, the manuscript can be significantly improved in terms of scientific novelty if more illustrated comparison and analysis of the presented results with previous will be given and other possible synthetic approaches to the target products will be shown or discussed. The manuscript can be suitable for publication in Molecules after a major revision. Additional comments are given below:

1)      In many cases yields are low, the analysis of side products can be important for further development. Is aldehyde group in starting compound and product stable (intact) under the reaction conditions?

2)      The manuscript should be carefully checked for misprints: “Cham-Lam” (should be Chan-Lam in the title), “crushed iced”, “After time completion NaBH4 (0.48 611 mmol) was added slowly and stirred for 20 min.” (please, rephrase to improve grammar), “the product extracted with EtOAc and purified” (“was” should be added before extracted)

3)      ”The reason for starting with these conditions and not with the conditions reported previously for the N-arylation of quinolones is that the electron donating effect of the methyl group at the p-position in substrate 8a can be beneficial for standardization of the protocol.” – the sentence in general is unclear, please, rephrase. It is not clear what previously reported conditions are meant here, reference should be added.

4)      The manuscript should be checked for self-consistence. For example: 1) temperature 80 °C is given in tables, 80-85 °C is given in experimental part, 2) Yield 82% is reported for 11 in Scheme 3, yield 83% is reported in experimental part

5)      In the preparation of 9a, experimental part: “the solution was stirred and NaOH (sol. 10 % w/v) added dropwise” the amount of NaOH should be specified.

6)      Additional references should be added about the synthesis and importance of quinoline derivatives as biologically active compounds:
(a) Xu, M.; Wagerle, T.; Long, J. K.; Lahm, G. P.; Barry, J. D.; Smith, R. M. Insecticidal Quinoline and Isoquinoline Isoxazolines. Bioorganic & Medicinal Chemistry Letters 2014, 24 (16), 4026–4030. https://doi.org/10.1016/j.bmcl.2014.06.004.

(b) Kaur, P.; Anuradha; Chandra, A.; Tanwar, T.; Sahu, S. K.; Mittal, A. Emerging Quinoline‐ and Quinolone‐based Antibiotics in the Light of Epidemics. Chem Biol Drug Des 2022, cbdd.14025. https://doi.org/10.1111/cbdd.14025.

(c) Vil’, V. A.; Grishin, S. S.; Baberkina, E. P.; Alekseenko, A. L.; Glinushkin, A. P.; Kovalenko, A. E.; Terent’ev, A. O. Electrochemical Synthesis of Tetrahydroquinolines from Imines and Cyclic Ethers via Oxidation/Aza‐Diels‐Alder Cycloaddition. Adv Synth Catal 2022, 364 (6), 1098–1108. https://doi.org/10.1002/adsc.202101355.

(d) Matada, B. S.; Pattanashettar, R.; Yernale, N. G. A Comprehensive Review on the Biological Interest of Quinoline and Its Derivatives. Bioorganic & Medicinal Chemistry 2021, 32, 115973. https://doi.org/10.1016/j.bmc.2020.115973.

Author Response

Query 1).  In many cases yields are low, the analysis of side products can be important for further development. Is aldehyde group in starting compound and product stable (intact) under the reaction conditions?

R/ One of the objectives of this work was to observe the stability of the aldehyde group, since it is fundamental for carrying out the structural modification of the N-arylated product. For this reason, the follow-up of the reaction was determined by NMR, thanks to this it was evidenced that this group remains intact during the reaction. On the other hand, the low yields are attributed to the homocoupling of the phenylboronic acids (formation of biphenyls) during the reaction.

Query 2). The manuscript should be carefully checked for misprints: “Cham-Lam” (should be Chan-Lam in the title), “crushed iced”, “After time completion NaBH4 (0.48 611 mmol) was added slowly and stirred for 20 min.” (please, rephrase to improve grammar), “the product extracted with EtOAc and purified” (“was” should be added before extracted)

R/ In line 3 the correction was made “Cham-Lam” (should be Chan-Lam in the title).
In the lines 327-333 the correction was made “crushed iced”. 

In the line 631-638 the correction was made “After time completion NaBH4 (0.48 611 mmol) was added slowly and stirred for 20 min.” (please, rephrase to improve grammar), “the product extracted with EtOAc and purified” (“was” should be added before extracted)

Query 3). The reason for starting with these conditions and not with the conditions reported previously for the N-arylation of quinolones is that the electron donating effect of the methyl group at the p-position in substrate 8a can be beneficial for standardization of the protocol.” – the sentence in general is unclear, please, rephrase. It is not clear what previously reported conditions are meant here, reference should be added.

R/ In line 122-125 the correction was made

Query 4). The manuscript should be checked for self-consistence. For example: 1) temperature 80 °C is given in tables, 80-85 °C is given in experimental part, 2) Yield 82% is reported for 11 in Scheme 3, yield 83% is reported in experimental part

R/ 1) In line 364 the correction was made

2)  In the scheme 3 the correction was made

Query 5). In the preparation of 9a, experimental part: “the solution was stirred and NaOH (sol. 10 % w/v) added dropwise” the amount of NaOH should be specified.

R/ In line 648 the correction was made

Query 6). Additional references should be added about the synthesis and importance of quinoline derivatives as biologically active compounds:
(a) Xu, M.; Wagerle, T.; Long, J. K.; Lahm, G. P.; Barry, J. D.; Smith, R. M. Insecticidal Quinoline and Isoquinoline Isoxazolines. Bioorganic & Medicinal Chemistry Letters 2014, 24 (16), 4026–4030. https://doi.org/10.1016/j.bmcl.2014.06.004.

(b) Kaur, P.; Anuradha; Chandra, A.; Tanwar, T.; Sahu, S. K.; Mittal, A. Emerging Quinoline‐ and Quinolone‐based Antibiotics in the Light of Epidemics. Chem Biol Drug Des 2022, cbdd.14025. https://doi.org/10.1111/cbdd.14025.

(c) Vil’, V. A.; Grishin, S. S.; Baberkina, E. P.; Alekseenko, A. L.; Glinushkin, A. P.; Kovalenko, A. E.; Terent’ev, A. O. Electrochemical Synthesis of Tetrahydroquinolines from Imines and Cyclic Ethers via Oxidation/Aza‐Diels‐Alder Cycloaddition. Adv Synth Catal 2022, 364 (6), 1098–1108. https://doi.org/10.1002/adsc.202101355.

(d) Matada, B. S.; Pattanashettar, R.; Yernale, N. G. A Comprehensive Review on the Biological Interest of Quinoline and Its Derivatives. Bioorganic & Medicinal Chemistry 2021, 32, 115973. https://doi.org/10.1016/j.bmc.2020.115973.

R/ the references were added in lines 48-50

Reviewer 2 Report

In this manuscript, the author describes a copper-catalyzed Cham-Lam coupling using 3-formylquinolin-2(1H)-one as substrate. Although there are plenty relevant precedents of quinolin-2(1H)-one derivatives, the C-N coupling from -CHO contained quinolin-2(1H)-one substrate has not been established yet, and the further transformation of the -CHO contained product is of great interests, thus the manuscript has its novelty and scientific significance. However, the yield is not good, some of them even lower than 10%, which is unacceptable for a research article of methodology development. Besides, there are many small errors in this manuscript. In summary, this reviewer consider the results are useful but a major revision is essential.

(1) Line 17, in the abstract, the author claims 32 compounds are synthesized. However, 5 of them (9g, 9h, 9i, 9j, 9k) have extremely low yield (lower than 10%), which is unacceptable for the research of methodology development, the author should try to enhance the yield or place these examples from Table 2 to Supplementary Information, with labelling “low efficient transformation”.

(2) In the section of Table 1, if the yield can be further improved by increasing the amount of catalyst loading to 20 mol%?

(3) Line 231, Table 2, the solvent volume of 1mmol scale is 10 ml, but in line 343, the solvent volume of 1mmol scale is 12.5 ml, which one is correct? Besides, the volume of solvent is too much, almost 60 V compared to 7a, why did the author add so many solvent and what would happen if decrease the solvent volume to 20 V (1mmol 7a ~ 3.5 ml DMF)?

(4) All of the presentations of catalyst loading must be consistent (Scheme 1: CuOTf, line 126, Table 1, etc.). “10 mol%” should be a standard example.

(5) The author should carefully check the compound number. In line 19, the author refers to the transformations of 9a, but in Scheme 3, several transformations are based on 9b; Similarly, in line 167 and 174, compound 9b is mentioned, but the structure in Fig 1 is 9a as per Table 2.

(6) Scheme 1, in the section of “Previous work”, pathway b and c, the R and R’ group are not consistent.

(7) Table 1, Entry 11, CsCO3 should be Cs2CO3.

(8) Line 280-285, Scheme 3, “0,3” should be “0.3”, and the starting material should be labelled as 9b, as per Table 2.

Author Response

Query 1). Line 17, in the abstract, the author claims 32 compounds are synthesized. However, 5 of them (9g, 9h, 9i, 9j, 9k) have extremely low yield (lower than 10%), which is unacceptable for the research of methodology development, the author should try to enhance the yield or place these examples from Table 2 to Supplementary Information, with labelling “low efficient transformation”.

R/ The observations made by the referee were met, so lines 187-198 detail the process used to improve the yield of products 9g, 9h, 9i, 9j, 9k

Query 2). In the section of Table 1, if the yield can be further improved by increasing the amount of catalyst loading to 20 mol%?

R/ In Table 1, entry 17 was added and the yield did not improve with the increase of catalyst loading.

Query 3). Line 231, Table 2, the solvent volume of 1mmol scale is 10 ml, but in line 343, the solvent volume of 1mmol scale is 12.5 ml, which one is correct? Besides, the volume of solvent is too much, almost 60 V compared to 7a, why did the author add so many solvents and what would happen if decrease the solvent volume to 20 V (1mmol 7a ~ 3.5 ml DMF)?

R/ The correct amount of solvent is 10 mL (line 278). The reason for adding this amount, is due to the low solubility of 3-formylquinolin-2(1H)-ones

Query 4). All of the presentations of catalyst loading must be consistent (Scheme 1: CuOTf, line 126, Table 1, etc.). “10 mol%” should be a standard example.

R/ Based on the observation of the referee, the correction was made in scheme 1

Query 5) The author should carefully check the compound number. In line 19, the author refers to the transformations of 9a, but in Scheme 3, several transformations are based on 9b; Similarly, in line 167 and 174, compound 9b is mentioned, but the structure in Fig 1 is 9a as per Table 2.

R/ In the scheme 3 the correction was made “The author should carefully check the compound number. In line 19, the author refers to the transformations of 9a, but in Scheme 3, several transformations are based on 9b”.

The observation of the referee regarding lines 163 and 170 was made in lines 159 and 166

Query 6) Scheme 1, in the section of “Previous work”, pathway b and c, the R and R’ group are not consistent.

R/ Scheme 1 was corrected according to the observations made by the referee

Query 7). Table 1, Entry 11, CsCO3 should be Cs2CO3.

R/ The correction was made Table 1, Entry 11, Cs2CO3

Query 8). Line 280-285, Scheme 3, “0,3” should be “0.3”, and the starting material should be labelled as 9b, as per Table 2.

R/ The correction in scheme 3 was made according to the comments of the referee (line 301), it is clarified that the starting material is compound 9a (correction scheme 3)

Reviewer 3 Report

In the present manuscript Insuasty et al report on Cu(II)-catalyzed arylation of 3-formyl-2-quinolones using phenylboronic acids as arylating reagents. Although the reaction yields are not very favorable (5-64%), the advantage of their methodology is generation of N-arylquinolone derivatives under mild conditions (80C, pyridine as mild base) with cheap and readily available catalyst (CuO(OAc)2), and in an open flask (no need for inert atmosphere). In addition, their protocol tolerates a formyl group on a quinolone nucleus. They also discussed electronic effects of phenylboronic acids in the arylation process, and it seems that electron-donating groups favor higher yields than electron-withdrawing groups. The synthetic versatility of the obtained N-aryl-3-formyl-2-quinolones was nicely demonstrated by transforming a formyl group into other functionalities, the entire experimental work is carefully done and supports their conclusions. Products are adequately characterized, and the structures supported by spectroscopic methods. In my opinion, this work can be published in Molecules after some revision.

1) since the yields are in most cases below 50%, I wonder if the authors detected some by-products or were there only product+starting compounds in the reaction mixture after a given time? If yes, which? If not, why did they not prolong the reaction time?

2) Abstract: line 17: change …N-aryl-3-formyl-2-quinolones derivatives… into N-aryl-3-formyl-2-quinolone derivatives

Line 19: ‘’p’’ in ‘’p-methyl’’ should be italic; ‘’9a’’ should be bold

3) Introduction part: change Scheme 1 and rewrite the corresponding text (lines 51-58). I wonder why three products 1-3; they all have the same structure. There are also some inconsistencies, for ex., under product 4 we can find how R1, R2 and R’ are defined, but in the structure 4, we cannot see neither R2 nor R’….and there are some more as well as in the corresponding text, for ex., line 53 and 54: As stated, the product 2 is derived from arylation of 4,7-dimethyl-2-quinolone, but the structure does not support this. Simplify the footnote ‘’reagents and conditions’’.

4) Line 67: either you keep ‘’substituents’’ or ‘’groups’’, delete one of them

5) Scheme 1, this work: in the phenyl group of the product should be R’

6) Line 131: I understand that substrate is quinolone 7a and reagent boronic acid 8a. Please change ‘’substrate 8a’’ into ‘’reagent 8a’’.

7) Line 137:  instead of ‘’…of the structural scaffold 7a…’’ write ‘’…of the compound 7a…’’

8) Line 153: instead of ‘’…alkaline substances...’’ write ‘’…bases…’’

9) Lines 181-188: Conversion or yield? Yield is more important than conversion!

10) Rewrite lines 191-194

11) Line 203: m-OCF3 and p-OCF3, not only m- and p-!

12) Line 213: ‘’phenylboronic acids’’ not only ‘’boronic’’

13) Line 235: write ‘’para position’’ instead of ‘’p-position’’

14) Last paragraph, lines 266-276: text is not consistent with scheme 3; for ex., product 11 is not chalcone! Watch the yields! There is benzylamine in the scheme, whereas it should be p-methoxybenzylamine! And some more, have a look! Check also the footnote.

15) Experimental part  3.2.4. should be in consistence with Scheme 3

16) Conclusion:  line 636: omit ‘’a new’’. Line 643: instead of ‘’fitness’’ write ‘’potential’’

Author Response

Query 1). since the yields are in most cases below 50%, I wonder if the authors detected some by-products or were there only product+starting compounds in the reaction mixture after a given time? If yes, which? If not, why did they not prolong the reaction time?

R/ Homocoupling (biphenyls) products are generated by phenylboronic acid. Therefore, phenylboronic acid is depleted faster than the 3-formylquinolin-2(1H)-one nucleus. To avoid this problem, we decided to add the phenylboronic acids and the base in portions (as explained in the lines 187-198)

Now, the reaction was left for 48 h, but it showed no improvement in reaction yield (table 1, entry 16).

Query 2). Abstract: line 17: change …N-aryl-3-formyl-2-quinolones derivatives… into N-aryl-3-formyl-2-quinolone derivatives. Line 19: ‘’p’’ in ‘’p-methyl’’ should be italic; ‘’9a’’ should be bold

R/ In line 17 the correction was made “N-aryl-3-formyl-2-quinolones derivatives… into N-aryl-3-formyl-2-quinolone derivatives”. In line 19 the correction was made: ‘’p’’ in ‘’p-methyl’’ should be italic; ‘’9a’’ should be bold

Query 3). Introduction part: change Scheme 1 and rewrite the corresponding text (lines 51-58). I wonder why three products 1-3; they all have the same structure. There are also some inconsistencies, for ex., under product 4 we can find how R1, R2 and R’ are defined, but in the structure 4, we cannot see neither R2 nor R’….and there are some more as well as in the corresponding text, for ex., line 53 and 54: As stated, the product 2 is derived from arylation of 4,7-dimethyl-2-quinolone, but the structure does not support this. Simplify the footnote ‘’reagents and conditions’’.

R/ All the corrections indicated by the referee in scheme 1 were made, likewise, lines 57 and 64

Query 4) Line 67: either you keep ‘’substituents’’ or ‘’groups’’, delete one of them

R/ The correction was made on line 75 by keeping substituents

Query 5) Scheme 1, this work: in the phenyl group of the product should be R’

R/ All the corrections indicated by the referee in scheme 1 were made

Query 6) Line 131: I understand that substrate is quinolone 7a and reagent boronic acid 8a. Please change ‘’substrate 8a’’ into ‘’reagent 8a’’.

R/ The correction was made on line 122 according to the observation of the referee

Query 7) Line 137:  instead of ‘’…of the structural scaffold 7a…’’ write ‘’…of the compound 7a…’’

R/ The correction was made on line 129 according to the observation of the referee

Query 8) Line 153: instead of ‘’…alkaline substances...’’ write ‘’…bases…’’

R/ The correction was made on line 145 according to the observation of the referee

Query 9) Lines 181-188: Conversion or yield? Yield is more important than conversion!

R/ The correction was made on line 179 and 181 according to the observation of the referee

Query 10) Rewrite lines 191-194

R/ The lines 191-194 were rewritten by lines 187-198

Query 11) Line 203: m-OCF3 and p-OCF3, not only m- and p-!

R/ The correction was made on line 207

Query 12) Line 213: ‘’phenylboronic acids’’ not only ‘’boronic’’

R/ The correction was made on line 222

Query 13) Line 235: write ‘’para position’’ instead of ‘’p-position’’

R/ The corresponding correction was made in the line 238

Query 14) Last paragraph, lines 266-276: text is not consistent with scheme 3; for ex., product 11 is not chalcone! Watch the yields! There is benzylamine in the scheme, whereas it should be p-methoxybenzylamine! And some more, have a look! Check also the footnote.

R/ All the observations made by the referee in scheme 3 were made, likewise in the footnote (lines 300-303)

Query 15) Experimental part 3.2.4. should be in consistence with Scheme 3

R/ The corrections were made in scheme 3 and section 3.2.4 (Lines 601-603)

Query 16) Conclusion:  line 636: omit ‘’a new’’. Line 643: instead of ‘’fitness’’ write ‘’potential’’

R/ Corrections were made on lines 658 and 665

Round 2

Reviewer 1 Report

The line numbers Authors mention in their answers are not visible in the manuscript, for example, lines 364, 48-50. Thus it is hard to see the corrections in these cases. I also recommend to attach not highlighted version of the corrected manuscript for convenience of reading.

"To improve the yields shown by products 9g-9k, we opted to analyze the reaction mixture for these products, and it was clear that the main side product is the homocoupling of the corresponding phenylboronic acids. Taking this into account, we decided to extend the reaction time to 48 h obtaining 12, 11, 16, 14 and 18%, for 9g, 9h, 9i, 9j and 9k, respectively. Because the yields for products 9g, 9h, and 9j continued to be low ..." - The logical connection between first two sentences is broken. If the problem is the side process (not low conversion), then there is no reason to  prolongate the reaction time without the addition of additional portions of boronic acids.

Author Response

Query 1) The line numbers Authors mention in their answers are not visible in the manuscript, for example, lines 364, 48-50. Thus, it is hard to see the corrections in these cases.

R/ The references were added in lines 48-50 : a) insecticidal [20]; b) antibacterial [21]; c) antifungal [22], d) anti-inflammatory [27], among others [28–30].

Line 364, the correction was made: (open flask) at 80°C for 24 h.

Query 2). In line 186-195 the correction was made:  Subsequently, different halogenated phenylboronic acids replaced at the meta and para positions were used. Starting from the meta-F, Cl, Br and CF3 substituted precur-sors, the fluoro derivative 9f (15%) was obtained with a yield higher than 10%; unfor-tunately, meta substituted derivatives with Cl, Br, I, CF3 and OCF3 did not appear in yields greater than 10%. To improve the yields of 9g-9k, we opted to analyze the reac-tion mixtures and found that the main side products were the homocoupled phenyl-boronic acids. Taking this into account, we decided to add the base and the corre-sponding phenylboronic acids in portions (0.4 equivalents of each), separated by 90 minutes over a reaction time of 24 h. Using this methodology for the least satisfactory cases, the yields of the products 9g, 9h, and 9j rose to 45, 41 and 34%, respectively.

Reviewer 2 Report

The manuscript has been substantially improved, this referee believe the current version is suitable for publication.

Author Response

Thank you very much